# Plasticity in Classical Hodgkin Composite Lymphomas: A Systematic Review

**DOI:** 10.3390/cancers14225695

**Published:** 2022-11-19

**Authors:** Alexis Trecourt, Marie Donzel, Juliette Fontaine, Hervé Ghesquières, Laurent Jallade, Gabriel Antherieu, Camille Laurent, Claire Mauduit, Alexsandra Traverse-Glehen

**Affiliations:** 1Service de Pathologie Multi-Site, Site Sud, Centre Hospitalier Lyon Sud, Hospices Civils de Lyon, 69310 Pierre-Bénite, France; 2Faculté de Médecine Lyon-Sud, Université Claude Bernard Lyon 1, UR 3738—CICLY, 69921 Oullins, France; 3Faculté de Médecine de Lyon, Université Claude Bernard Lyon 1, 69100 Villeurbanne, France; 4Faculté de Médecine Lyon-Sud, CRCL, Centre International de Recherche en Infectiologie (CIRI), Université Claude Bernard Lyon-1, INSERM U1111, CNRS, UMR5308, ENS Lyon, 69921 Oullins, France; 5Service d’Hématologie Clinique, Centre Hospitalier Lyon Sud, Hospices Civils de Lyon, 69310 Pierre-Bénite, France; 6Laboratoire d’Hématologie, Centre Hospitalier Lyon Sud, Hospices Civils de Lyon, 69310 Pierre-Bénite, France; 7Service de Pathologie, Centre de Recherche en Cancérologie de Toulouse-Purpan, Institut Universitaire du Cancer, Oncopole de Toulouse, 31100 Toulouse, France; 8Institut National de la Santé et de la Recherche Médicale, Centre Méditerranéen de Médecine Moléculaire (C3M), Unité 1065, Equipe 10, 06000 Nice, France

**Keywords:** composite lymphoma, synchronous lymphoma, classical Hodgkin lymphoma, lymphomagenesis, transdifferentiation, plasticity

## Abstract

**Simple Summary:**

Composite/synchronous lymphoma is a rare entity, for which the histopathological diagnosis is difficult due to the co-occurrence of at least two lymphomas, sometimes mixed in the same anatomical site. In the present review, we gathered available data on composite lymphomas associating a classical Hodgkin lymphoma (cHL) with another lymphoma. We report the clinical, histopathological, immunohistochemical, and molecular data for each composite lymphoma. These data reinforce the hypothesis of a common clonal origin and a transdifferentiation phenomenon during lymphomagenesis. One of the greatest challenges for the pathologist is to differentiate real Hodgkin cells of cHL from Hodgkin-like cells associated with a non-Hodgkin lymphoma, in order to individualize both contingents for the diagnosis. In contrast, the clinician’s challenge is to optimally treat these rare composite pathologies as a single clinical entity. This review could thus be a useful diagnostic support for pathologists and could help clinicians improve management of these uncommon lymphomas.

**Abstract:**

The co-occurrence of several lymphomas in a patient defines composite/synchronous lymphoma. A common cellular origin has been reported for both contingents of such entities. In the present review, we aimed to gather the available data on composite lymphomas associating a classical Hodgkin lymphoma (cHL) with another lymphoma, to better understand the plasticity of mature B and T-cells. This review highlights that >70% of patients with a composite lymphoma are ≥55 years old, with a male predominance. The most reported associations are cHL with follicular lymphoma or diffuse large B-cell lymphoma, with over 130 cases reported. The cHL contingent is often of mixed cellularity type, with a more frequent focal/weak CD20 expression (30% to 55.6%) compared to de novo cHL, suggesting a particular pathophysiology. Moreover, Hodgkin cells may express specific markers of the associated lymphoma (e.g., BCL2/BCL6 for follicular lymphoma and Cyclin D1 for mantle cell lymphoma), sometimes combined with common *BCL2/BCL6* or *CCND1* rearrangements, respectively. In addition, both contingents may share similar *IgH*/*IgK* rearrangements and identical pathogenic variants, reinforcing the hypothesis of a common clonal origin. Finally, cHL appears to be endowed with a greater plasticity than previously thought, supporting a common clonal origin and a transdifferentiation process during lymphomagenesis of composite lymphomas.

## 1. Literature Review Section

### Introduction

A composite lymphoma is defined by the presence of at least two lymphomas in the same anatomical site [1]. However, the co-occurrence of several lymphomas in distinct locations, at the same time, is also accepted to qualify composite/synchronous lymphomas [2,3], as initially defined by Custer et al. in 1954 [4]. Indeed, synchronous and composite lymphomas might share the same pathophysiology, since common cellular origins have been identified between both contingents of these entities [5]. Due to the simultaneous occurrence of different lymphomas, these pathologies are interesting prototypes for studying the cellular plasticity that occurs during lymphomagenesis.

Among composite lymphomas, those associating a classical Hodgkin lymphoma (cHL) contingent have been of great interest since the beginning of the 21st century. Initially considered to be a coincidental association of two different morphological pathologies [2], common molecular abnormalities have been identified between both contingents of these lymphomas. Moreover, studies have also reported (i) common gene translocations (*BCL2, BCL6, CCND1*) [6,7,8,9,10]; (ii) common *IgH* and/or *IgK* rearrangements encoding the heavy and light immunoglobulin chains, respectively [10,11,12,13]; and (iii) common pathogenic variants (*TP53, BCL2, EP300, BCOR, KMT2D, ARID1A, SF3B1*) [8,10] between both contingents. To date, cHL have been reported in association with nodular lymphocyte-predominant Hodgkin lymphomas (NLPHL), B-cell lymphomas (follicular lymphomas, mantle cell lymphomas, marginal zone lymphomas, diffuse large B-cell lymphomas), and T-cell lymphomas [5]. The clinical behavior of cHL/B-cell composite lymphomas appears to be more indolent than that of cHL/B-cell sequential lymphomas, which are defined as several lymphomas arising consecutively in a patient [10,14,15,16]. However, the distinction between composite and sequential lymphomas is not always obvious in clinical practice. Although the lymphomagenesis process of composite lymphomas seems to differ from that of de novo cHL [17], this process remains to be deciphered. As previously suggested, the plasticity of mature B-cells may be responsible for a transdifferentiation phenomenon in the case of cHL/follicular composite lymphomas [10].

However, the main data reported on composite lymphomas are only based on case reports and small series, preventing a comprehensive clinical, histopathological, molecular, and pathophysiological study. In this review, composite lymphoma cases associating a cHL with another B or T-cell lymphoma were gathered from the literature, in order to increase understanding of cHL plasticity.

## 2. Materials and Methods

The data collection was performed from July to August 2022, on the PubMed Central (PMC) archives (NCBI), using the keywords “composite lymphoma(s)”, “synchronous lymphoma(s)”, “composite AND Hodgkin”, “synchronous AND Hodgkin”, without limit regarding the year of publication (from 1977 to 2022). One reviewer (A.T.) sorted all the articles found, to perform a first selection of those meeting the inclusion criteria. In addition, an exhaustive study of the reference section of each article was carried out. Then, all articles selected were checked independently by another reviewer (A.T.-G.).

Inclusion criteria were (i) the co-occurrence of a cHL with other lymphoma(s) diagnosed at the same time, regardless of their location or the patient’s history; and (ii) the presence of two or more distinct, well-defined, and histopathologically proven non-cutaneous lymphomas. Exclusion criteria were (i) NLPHL or non-Hodgkin lymphoma (NHL) without a cHL contingent associated; (ii) absence of simultaneous occurrence (sequential lymphoma); (iii) association of cHL and chronic lymphocytic leukemia (CLL), which is known as a transformation process rather than a composite pathology; (iv) the presence of Hodgkin-like cells occurring in an NHL; and (v) report(s) including the diagnosis of cHL with negative CD30 “Hodgkin cells”. After the final selection, all articles included were separated into six groups: composite lymphoma associating a cHL and either: (i) NLPHL; (ii) follicular lymphoma; (iii) mantle cell lymphoma; (iv) marginal zone lymphoma; (v) diffuse large B-cell lymphoma (DLBCL); or (vi) T-cell lymphoma. The clinical, laboratory, histopathological, and molecular data from each article available in English were collected. As the chemotherapy treatment that was used was different from one study to another, lymphoma treatments were classified into four groups: cHL-like, B-cell lymphoma-like, T-cell lymphoma-like, and composite lymphoma-like chemotherapy. The latter included combinations of cHL-like chemotherapy and another lymphoma treatment (see Appendix A for the classification of treatments). Of note, for the cHL/follicular composite lymphomas previously published by our group [10], more details have been added herein due to the ongoing follow-up of these patients in our center in Lyon, France.

The results of all included studies have been compiled in tables (see Appendix A). Then, the frequencies of the variables from all studies were described as percentages and medians for dichotomous and continuous values, respectively.

Microscopic photographs were obtained using Aperio ImageScope software v12.3.3 (Leica Biosystems, Nussloch, Germany) after slide scanning using Aperio AT2 scanner (Leica Biosystems). Figures and Appendix A were created using BioRender (BioRender, Toronto, Canada; BioRender.com). All the figures included in the present review (main text and Appendix A) are original.

This systematic review was written according to the PRISMA 2020 Checklist (prisma-statement.org; see Appendix A).

## 3. Results

### 3.1. Article Selection Process for Final Analysis

More than 700 articles were retrieved in PubMed. After an initial screening, 108 articles, published from 1977 to 2022, were identified as “composite/synchronous lymphomas with a Hodgkin lymphoma”. Among them, nine articles were not included: (i) one because both contingents (cHL/NLPHL) had the same immunohistochemical phenotype and were difficult to individualize [18]; (ii) eight because the second contingent was a cutaneous lymphoproliferation (T-cell lymphoma, mycosis fungoid, or lymphomatoid papulosis) [19,20,21,22,23,24,25,26].

Among the remaining 99 articles, 26 were excluded from the final analysis because they met at least one exclusion criterion: (i) six NLPHL/NHL composite lymphomas without cHL contingent [27,28,29,30,31,32]; (ii) two cHL with an Epstein–Barr virus (EBV) post-transplant lymphoproliferative disorder [33] or EBV mucocutaneous ulcer [34]; (iii) one with CD30 negative Hodgkin cells [35]; (iv) four with Hodgkin-like cells within an NHL [36,37,38,39]; (v) two sequential lymphomas [40,41]; (vi) ten probable transformations of CCL into cHL [42,43,44,45,46,47,48,49,50,51]; and (vii) one because the final diagnosis could be considered ambiguous [52].

As a result, 73 original articles reporting composite lymphomas with a cHL contingent were included in the final analysis. The flow diagram of article inclusion/exclusion is presented in Appendix A [53].

### 3.2. cHL and Follicular Composite Lymphomas

A total of 76 cHL/follicular composite lymphomas from 20 articles were analyzed [3,6,10,11,12,14,15,54,55,56,57,58,59,60,61,62,63,64,65,66].

#### 3.2.1. Clinical and Laboratory Data

The median age was 59.5 years (ranging from 35 to 84), and the male/female ratio was 2. At presentation, 59.2% (45/76) of patients had no history of lymphoma, 1.3% (1/76) had a previous history of cHL, 7.9% (6/76) had a previous history of follicular lymphoma, associated with a DLBCL in one case. The history of lymphoma was not specified in 32.9% (25/76) of cases. The median follow-up time was 23.5 months (ranging from 0.75 to 171 months). Fourteen patients relapsed after remission: five as follicular lymphoma, three as cHL, and six without other specification. For patients who relapsed/died from lymphoma, the median relapse time was 14 months (ranging from 1 to 108 months). Clinical and laboratory data are presented in Table 1 and Table 2, and Appendix A.

#### 3.2.2. Pathological Data

The cHL contingent was of mixed cellularity-type (MC) or with an abundant granulomatous background in 60.5% (46/76) of cases, nodular sclerosis-type (NS) in 23.7% (18/76), and unclassified in 15.8% (12/76) of cases. The follicular lymphoma contingent was in situ in 1.3% (1/76), grade 1–2 in 53.9% (41/76), grade 3A in 13.2% (10/76), grade 3B in 2.6% (2/76), and unspecified in 28.9% (22/76) of cases. A DLBCL contingent was associated in 7.9% (6/76) of cases. The major contingent was follicular lymphoma and/or DLBCL in 28.9% (22/76), cHL in 13.2% (10/76), and unspecified in 53.9% (41/76) of cases. Both contingents were present in equal quantity in 3.9% (3/76) of cases. Pathological data are presented in Table 1 and Table 3, and Appendix A.

#### 3.2.3. Molecular Data

A *BCL2* translocation was reported in both contingents in 57.7% (15/26) of cases, and a translocation of *BCL6* was identified in 8.3% (1/12) of cases. In addition, a common 16p duplication was also reported in one case.

Similar clonal rearrangements of *IgH*/*IgK* were reported in 90% (9/10) of cases. In the only negative case, DNA degradation was reported.

Two cases of cHL/follicular composite lymphomas showed identical pathogenic variants of *BCL2, EP300, BCOR, ARID1A, KMT2D,* and *SF3B1* in both contingents obtained by microdissection. In addition, each contingent showed its own pathogenic variants, such as *BCL2, KMT2D, EP300* in the follicular lymphoma contingent, and *XPO1, TNFAIP3,* and *CREBBP* in the cHL contingent [10]. Molecular data are detailed in Appendix A.

### 3.3. cHL and Mantle Cell Composite Lymphomas

A total of 11 cHL/mantle cell composite lymphomas from 10 articles were analyzed [6,8,9,67,68,69,70,71,72,73].

#### 3.3.1. Clinical and Laboratory Data

The median age was 67.5 years (ranging from 42 to 89), and the male/female ratio was 4. At presentation, 45.5% (5/11) of patients had no history of lymphoma, 54.5% (6/11) had a history of lymphoma (mantle cell lymphoma in five cases and unspecified lymphoproliferation in one case). Data concerning the median follow-up time were not available in 90.9% (10/11) of cases. Clinical and laboratory data are presented in Table 1 and Table 2, and Appendix A.

#### 3.3.2. Pathological Data

The cHL contingent was of NS-type in 9.1% (1/11), MC-type in 9.1% (1/11), lymphocyte-rich-type (LR) in 18.2% (2/11), and unclassified in 63.6% (7/11) of cases. The mantle cell lymphoma contingents were of classical-type in 90.9% (10/11) and blastoid-type in 9.1% (1/11) of cases. Pathological data are presented in Table 1 and Table 3, and Appendix A.

#### 3.3.3. Molecular Data

A translocation of *CCND1* was reported in both contingents in 62.5% (5/8) of cases.

A similar clonal *IgH* rearrangement in both contingents was reported in 40% (2/5) of cases, while distinct *IgH* rearrangements were found in 60% (3/5) of cases. In one patient, a *TP53* variant (exon 5, p.Y163C) and a heterozygous deletion of *TP53*/*17p13* were identified. Molecular data are detailed in Appendix A.

### 3.4. cHL and Marginal Zone Composite Lymphomas

A total of 21 cHL/marginal zone composite lymphomas from 10 articles were analyzed [6,60,74,75,76,77,78,79,80,81].

#### 3.4.1. Clinical and Laboratory Data

The median age was 68.5 years (ranging from 45 to 87), and the male/female ratio was 4. Most patients (76.2%, 16/21) had no history of lymphoma, 14.3% (3/21) had a history of lymphoma (mucosa-associated lymphoid tissue lymphoma (MALT), splenic marginal zone lymphoma (SMZL), and lymphoplasmacytoid immunocytoma/lymphoma), and the history of lymphoma was not specified in 9.5% (2/21) of cases. The median follow-up time was 12 months (ranging from 2 to 131 months). For patients who relapsed/died from lymphoma, the median relapse time was 27.5 months (ranging from 7 to 122). Clinical and laboratory data are presented in Table 1 and Table 2, and Appendix A.

#### 3.4.2. Pathological Data

The cHL contingent was of NS-type in 4.8% (1/21), MC-type in 57.1% (12/21), and unclassified in 38.1% (8/21) of cases. The second contingent was SMZL, MALT, MALT + DLBCL, MZBL, and MZBL + DLBCL in 14.3% (3/21), 33.3% (7/21), 28.6% (6/21), 19% (4/21), and 4.8% (1/21) of patients, respectively. Pathological data are presented in Table 1 and Table 3, and Appendix A.

#### 3.4.3. Molecular Data

For the only case tested, no t(11;18) translocation and no *IgH* rearrangement was identified in either of the contingents individualized before testing. Molecular data are detailed in Appendix A.

### 3.5. cHL and Diffuse Large B-Cell Composite Lymphomas

A total of 79 cHL/diffuse large B-cell composite lymphomas from 35 articles were analyzed [1,3,6,13,14,16,58,64,66,76,81,82,83,84,85,86,87,88,89,90,91,92,93,94,95,96,97,98,99,100,101,102,103,104,105].

#### 3.5.1. Clinical and Laboratory Data

The median age was 58 years (ranging from 14 to 84), and the male/female ratio was 1.5. At presentation, 57% (45/79) of patients had no history of lymphoma, 6.3% (5/79) had a history of lymphoma (cHL in two cases, follicular lymphoma in one case, DLBCL in one case, DLBCL and follicular lymphoma in one case), and the history of lymphoma was not specified in 36.7% (29/79) of cases. The median time of follow-up was 12 months (ranging from 0.67 to 124.8 months). Five patients relapsed after complete remission: 60% (3/5) as cHL, 20% (1/5) as DLBCL, and 20% (1/5) had no available data. For patients who relapsed/died from lymphoma, the median relapse time was 7 months (ranging from 1 to 54 months). Clinical and laboratory data are presented in Table 1 and Table 2, and Appendix A.

#### 3.5.2. Pathological Data

The cHL contingent was of NS-type in 53.2% (42/79), MC-type in 21.5% (17/79), LR-type in 1.3% (1/79), and unclassified/unspecified in 24% (19/79) of cases. The second contingent was DLBCL in 70.9% (56/79), Burkitt leukemia/lymphoma in 1.3% (1/79), primary mediastinal B-cell lymphoma in 25.3% (20/79), gray-zone lymphoma in 1.3% (1/79), and high-grade B-cell lymphoma in 1.3% (1/79) of cases. Pathological data are presented in Table 1 and Table 3, and Appendix A.

#### 3.5.3. Molecular Data

Similar clonal *IgH*/*IgK* rearrangements were found in 70% (7/10) of cases. More specifically, in two cases, both identical and different *V* gene somatic hypermutations were identified in both contingents. In one case, somatic hypermutations of *IgVH* were increased in the DLBCL contingent (27–28%) compared to the cHL contingent (7–8%). In one case, a *TP53* variant was found only in the DLBCL contingent. Molecular data are detailed in Appendix A.

### 3.6. cHL and Nodular Lymphocyte-Predominant Hodgkin Composite Lymphomas

A total of three cHL/NLPHL composite lymphomas from three articles were analyzed [106,107,108].

#### 3.6.1. Clinical and Laboratory Data

The median age was 24 years (ranging from 22 to 48), and the three cases were male. At presentation, no patient had a personal history of lymphoma, but one patient had a family history of cHL (father and two paternal cousins). The follow-up time was available for only one patient (12 months). Clinical and laboratory data are presented in Table 1 and Table 2, and Appendix A.

#### 3.6.2. Pathological Data

The cHL contingent was of NS-type in 33.3% (1/3) and MC-type in 66.7% (2/3) of cases. MUM1, LMP-1, and p53 immunostainings were each performed in only one case, and found positive in the cHL contingent where they were performed. Similarly, CD79a, MUM1, BOB1, and p53 immunostainings were positive in the only case of NLPHL contingent where they were performed. Pathological data are presented in Table 1 and Table 3, and Appendix A.

#### 3.6.3. Molecular Data

In one case, an identical clonal *IgH* rearrangement with the same *V* gene somatic hypermutations were identified in both contingents. A second different *IgH* rearrangement was observed in the NLPHL contingent only. For another patient, the *IgH* repertoire was polyclonal, but both contingents shared a similar clonal *IgK* rearrangement. Molecular data are detailed in Appendix A.

### 3.7. cHL and T-cell Composite Lymphomas

A total of 11 cHL/T-cell composite lymphomas from seven articles were analyzed [60,109,110,111,112,113,114].

#### 3.7.1. Clinical and Laboratory Data

The median age was 57 years (ranging from 19 to 84), and the male/female ratio was 2.7. At presentation, 72.7% (8/11) of patients had no history of lymphoma and 27.3% (3/11) had a history of lymphoma/hematological pathology (a cHL in two cases, a prolymphocytic leukemia in one case). The median follow-up time was 30 months (ranging from 0.75 to 48 months). Clinical and laboratory data are presented in Table 1 and Table 2, and Appendix A.

#### 3.7.2. Pathological Data

The cHL contingent was of NS-type in 9.1% (1/11), MC-type in 45.5% (5/11), and unclassified in 45.5% (5/11) of cases. The second contingent was peripheral T-cell lymphoma in 72.7% (8/11), T-cell prolymphocytic leukemia in 18.2% (2/11), and cytotoxic CD8+ T-cell lymphoma in 9.1% (1/11) of cases.

Concerning the cHL contingent, no expression of T-cell markers (CD2, CD3, CD4, CD5, CD8), ALK-1, nor LCA/CD45 were observed in the Hodgkin cells. For the T-cell lymphoma contingent, the proliferative index (Ki67) ranged from 40% to 80%. The CD30 immunostaining of T-cell lymphomas was always negative. Concerning the cHL contingent, the LMP-1 immunostaining performed in only one case was positive, while the ALK-1, CD4, and CD8 immunostainings, also performed in only one case, were negative. For the T-cell lymphoma contingent, the CD20 immunostaining performed in only one case was negative. Pathological data are presented in Table 1 and Table 3, and Appendix A.

#### 3.7.3. Molecular Data

For two cases, a *TCR-β* and/or *TCR-γ* rearrangement was identified in both T-cell lymphoma contingents, while a clonal *IgH* rearrangement was found only in one cHL contingent.

For all the other cases tested, the rearrangements were analyzed on the whole tissue samples (non-individualized contingents). Clonal *IgH* rearrangements were identified in 83.3% (5/6) of cases, and clonal *TCR-β* and/or *TCR-γ* rearrangements were identified in 100% (9/9) of cases. No common clonal B or T rearrangement was reported in either of the individualized contingents of the cHL/T-cell composite lymphomas. Molecular data are detailed in Appendix A.

## 4. Discussion

The present systematic review, which gathered the available data from the literature concerning composite lymphomas with a cHL contingent, allowed to describe the clinical, histopathological, and molecular characteristics of these entities, to improve the understanding of lymphomagenesis in these rare lymphomas (Figure 1). Due to the low prevalence of these diseases (<0.5% of lymphomas) [115], no prospective study can be easily performed and the data available are thus based on case reports and small series only. This review highlights that >70% of patients with composite lymphomas are ≥55 years old, suggesting that aging and immunosenescence could be risk factors for developing two lymphomas simultaneously [68]. However, cHL/NLPHL [106,108], cHL/DLBCL [14,66,84,86,89], and cHL/T-cell lymphoma [109] were also reported in younger patients (<35 years old), emphasizing that these pathologies can occur at any age. Histologically, the cHL contingent is very frequently of MC-type, suggesting a particular pathophysiology. Moreover, Hodgkin cells within composite lymphomas often express CD20 in a focal and/or weak manner (from 28.9% to 55.6%), while this expression is reported to be less frequent in de novo cHL (from 12.6 to 28.1%) [116,117,118,119]. These Hodgkin cells may also express markers of the associated lymphoma such as BCL2/BCL6 for follicular lymphoma and Cyclin D1 for mantle cell lymphoma, which may be combined with *BCL2/BCL6* and *CCND1* rearrangements in both contingents, respectively [3,6,8,10,56,61,62,70,71,72]. These contingents may also share similar *IgH*/*IgK* rearrangements and identical pathogenic variants [5,10,56], reinforcing the hypothesis of a common clonal origin. Although the classification of lymphomas, and especially the distinction between germinal center and post-germinal center subgroups, is currently based on cellular origin [120], it appears that the cellular plasticity occurring during lymphomagenesis might be higher than initially thought. Follicular lymphomas, which have a germinal center origin [120], are an ideal illustration of this cellular plasticity, since they have been reported to transform into post-germinal center DLBCL [121,122] or transdifferentiate into myeloid pathologies such as histiocytic/dendritic cell sarcomas in rare instances [123]. The data presented herein thus expand the spectrum of mature B-cell plasticity and the classical limits of lymphomagenesis.

At diagnosis, the challenge is to differentiate a cHL contingent from Hodgkin-like cells, which can co-occur within B-cell [36,37] or T-cell lymphomas [124,125] (an example of such a differential diagnosis is given in Figure 2). Some criteria must be considered to recognize Hodgkin-like cells: (i) absence of fibrosis and/or cHL stromal reaction (eosinophils, histiocytes) with mixed/non-nodular distribution of Hodgkin cells [18,33,36,39,79]; (ii) absence/weak expression of usual cHL markers such as CD30 [35] or CD15 [36,59]; (iii) diffuse and intense expression of CD45 and multiple B-cell markers [37]. Moreover, since it is classically accepted that the lymphoma pathophysiology follows the lymphoma negation principle [120], the presence of reactive germinal centers and/or post-germinal center immunoblasts with a normal differentiation spectrum are not suggestive of cHL diagnosis [120]. However, the distinction is sometimes difficult to achieve in practice, and especially in the context of EBV lymphoproliferative disorders [33,34] in which the study of EBV latency could be helpful [126].

### 4.1. cHL/Follicular Composite Lymphomas

Patients are generally old or middle-aged and without a history of lymphoma. At presentation, a polyadenopathy and a III/IV Ann Arbor stage are observed. The cHL contingent is classically of MC-type, less frequently of NS-type, and is usually separated from the follicular lymphoma contingent, which is more often grade 1–2. Patients treated with a composite lymphoma-like chemotherapy showed a rate of relapse/death slightly decreased compared to other patients, reinforcing the importance of treating both contingents.

It has been hypothesized that the cHL contingent of composite cHL/follicular lymphomas could be different than de novo cHL [10,56]. In this review, we also observed (i) clinical differences (composite lymphomas occurring at a later age than de novo cHL); (ii) morphological differences (high prevalence of EBER negative cHL-MC compared to de novo cHL-MC which is often EBER positive, reinforcing the hypothesis of a low prevalence of EBV infection in composite lymphomas [66]); (iii) immunohistochemical differences with a higher prevalence of centrofollicular markers and some B-cell markers, questioning a possible response to anti-CD20 therapy of the cHL contingent [57]; and (iv) molecular differences, with *BCL2/BCL6* translocations and pathogenic follicular lymphoma-like variants reported in the cHL contingent [3,10,56,61,62]. All these uncommon features suggest a specific pathophysiology of these lymphomas and as shown in Figure 3, a transdifferentiation process of the cHL contingent from a follicular tumor cell could be hypothesized.

### 4.2. cHL/Mantle Cell Composite Lymphomas

This entity is rarer than cHL/follicular composite lymphomas, with only a few cases reported in the literature. Patients are classically old males with a history of mantle cell lymphoma, presenting polyadenopathy but without B symptoms. The Ann Arbor stage is usually III/IV, without bone marrow involvement. Both contingents are often mixed with frequent Hodgkin cell CD15 and EBER positivity. As shown in Figure 4, data available could support a transdifferentiation process of cHL from a tumor mantle cell precursor, with some evidence that the cHL contingent could then pass through the germinal center [6,9].

### 4.3. cHL/Marginal Zone Composite Lymphomas

Patients are classically old males without a history of lymphoma and presenting polyadenopathy but without B symptoms. At presentation, the Ann Arbor stage is classically III/IV. Both contingents are usually separated, with a cHL-MC contingent expressing p53 and often CD15, LMP-1, and EBER, while the marginal zone lymphoma contingent does not. The rate of relapse/death slightly decreased when patients were treated with small/diffuse B-cell lymphoma-like chemotherapy.

No common clonal origin has been identified between both contingents. However, the molecular data available are very limited with only one case reported without *IgH* rearrangement or somatic hypermutations [6]. In addition, the cellular origins are different as cHL pass through the germinal center while marginal and splenic marginal zone lymphomas pass outside the germinal center [127]. However, an activation of the NF-_K_B pathway is implicated in the lymphomagenesis of both neoplasms [128,129,130].

### 4.4. cHL/NLPHL Composite Lymphomas

This entity seems to be the less frequent of all composite lymphomas. Whether the presentation is simultaneous or sequential, the association of cHL and NLPHL is described especially in young males [108]. In one case, a familial history of cHL was reported [106]. Indeed, although the pathophysiology between cHL and NLPHL is different [17], it has been suggested that there is a greater risk of developing Hodgkin lymphoma in case of familial history of the same pathology [131], with a predominantly male predisposition [132] and a cumulative risk of Hodgkin lymphoma in first-degree relatives of 0.6% [133]. Recently, familial syndromes with Hodgkin lymphomas have been reported including germinal homozygote CD27 deficiency [134], DICER1 syndrome [36,135,136], association with the human leucocyte antigen (HLA) abnormalities [137], *KLHDC8B* translocation [138], *NPAT* germinal mutation [139], homologous germinal variant of *ACAN* [140], and familial *KDR* mutations [141]. However, the presence of certain homologous variants in healthy patients supports the necessity but the insufficiency of these genetic defects alone to cause the disease, with a potential role of environmental factors [134]. The exact lymphomagenesis of this neoplasm is unknown. However, based on the data available, we propose a model of lymphomagenesis in Figure 5.

### 4.5. cHL/DLBCL Composite Lymphomas

Patients are classically old or middle-aged, without history of lymphoma at presentation, and with polyadenopathy but without B symptoms. Both contingents are usually separated, with a cHL contingent of NS-type and less frequently MC-type. Hodgkin cells often express CD15 and p53 (without *TP53* mutation reported in the cHL contingent). These lymphomas have been reported with a better prognosis than those with a sequential presentation [16,142]. The rate of relapse/death slightly decreased when patients were treated with small/diffuse B-cell lymphoma-like chemotherapy. Based on the data available, we suggest a model of lymphomagenesis in Figure 6.

Recently, Singh et al. have reported a series of six cHL/DLBCL and DLBCL/cHL sequential lymphomas, and identified identical variants in both contingents (including *TNFAIP3, XPO1, TP53,* and *B2M*), reinforcing the clonal relationship between these lymphomas and the concept of plasticity occurring within mature B-cells [143].

### 4.6. Composite cHL/T-Cell Lymphomas

Patients are classically old or middle-aged, without history of lymphoma at presentation, and present polyadenopathy with or without B symptoms. The cHL contingent is more frequently an MC-type. Hodgkin cells often express p53 and CD15. In the T-cell lymphoma contingent, the less expressed T-cell markers are CD7 and then CD4. In contrast, CD2 and CD3 were always reported as positive on neoplastic T-cells.

To date, there is no molecular evidence of a clonal relationship between both contingents of these composite lymphomas, because, in most cases, the molecular studies were performed on the whole sample without individualization of the two contingents. In addition, no shared B or T rearrangements were observed in the only two cases for which the contingents were individualized [60,110]. However, despite the classic B-cell origin of cHL [127], some authors have identified a T-cell origin of certain cHL, with a clonal rearrangement of *TCR-γ* [144] or *TCR-β* [145] in Hodgkin cells. Additionally, common clonal rearrangements of the TCR-encoding gene seem to occur between both contingents of cHL/T-cell sequential lymphomas [112,146], but the histopathological differential diagnosis between cHL and T-cell lymphomas is sometimes difficult in practice. Moreover, cHL have been reported in association with other cutaneous T-cell neoplasms, such as mycosis fungoid or lymphomatoid papulosis [22,112,146], with or without TCR rearrangements, but these neoplasms were excluded in the present review because the pathophysiology is probably different.

Distinguishing between peripheral T-cell lymphoma with Hodgkin-like cells and LR-type cHL or cHL/T-cell composite lymphoma could be very misleading in practice [147,148]. Moreover, the diagnosis of T-cell lymphoma with Hodgkin-like cells is probably more frequent than that of cHL/T-cell composite lymphoma. These Hodgkin-like cells could be EBV positive or EBV negative (especially in T-cell lymphomas of TFH origin), and their presence is commonly interpreted as a consequence of the patient’s immunosuppression [147,148]. The presence of aggregates or sheets of atypical T-cells with clear cytoplasm, expressing CD10 and CXCL13, must alert the pathologist of a potential diagnosis of T-cell lymphoma rather than cHL. Indeed, other TFH-associated markers such as BCL6, PD1, ICOS, are less useful as they are also commonly observed in the microenvironment of cHL [147,148]. Other criteria such as (i) the absence of fibrosis/sclerosis and polymorphic background; (ii) the presence of Hodgkin-like cells distributed in a background of neoplastic TFH-cells without distinct areas of cHL; (iii) the absence of the single layer of rosetting T-cells surrounding the Hodgkin-like cells, replaced by a large aggregate of T cells; and (iv) a more preserved B-cell program of the Hodgkin-like cells with diffuse and intense expression of B-cell markers, also support the diagnosis of T-cell lymphoma with Hodgkin-like cells rather than that of cHL [147,148].

### 4.7. Limits and Conclusions

This review has some limits. First, the clinical data available are very limited, since most articles are unique cases that do not report the patient’s follow-up and/or treatments. In addition, patient management often differs between studies, even within the same type of composite lymphoma, complicating their comparison and the establishment of treatment guidelines. As a result, the clinical conclusions from these studies are less robust. This is particularly true for laboratory data or the extension assessment, for which data were often missing. However, in the absence of prospective studies, retrospective data remain the only way to understand these rare lymphomas. Another limitation relates to the methodological biases of the review: (i) the non-exhaustive nature of the PubMed database; (ii) the initial selection of articles made by only one reviewer (A.T.); and (iii) the article selection which was dependent on the keywords used. However, to reduce some of these biases, an exhaustive study of the bibliography/references of each article included in this review was also carried out, in order to search for articles that would not have been found using the keywords previously cited. In addition, a cross-check of all the articles finally included was performed independently by a second reviewer (A.T.-G.). Moreover, the objective of this review was not to exhaustively describe all clinical, histopathological, and molecular data reported for composite lymphomas, but to estimate the frequency of the variables available in medical-scientific articles in English for composite lymphomas, and to better understand the cellular plasticity occurring in cHL (Figure 3, Figure 4, Figure 5 and Figure 6).

## 5. Conclusions

To conclude, cHL appears to be endowed with a greater plasticity spectrum than previously thought, explaining its association with multiple T and B lymphoid neoplasms, or even myeloid disorders [149]. The lymphomagenesis of composite lymphomas is complex and probably multifactorial, combining genetic, infectious, and environmental factors.

## Figures and Tables

**Figure 1 cancers-14-05695-f001:**
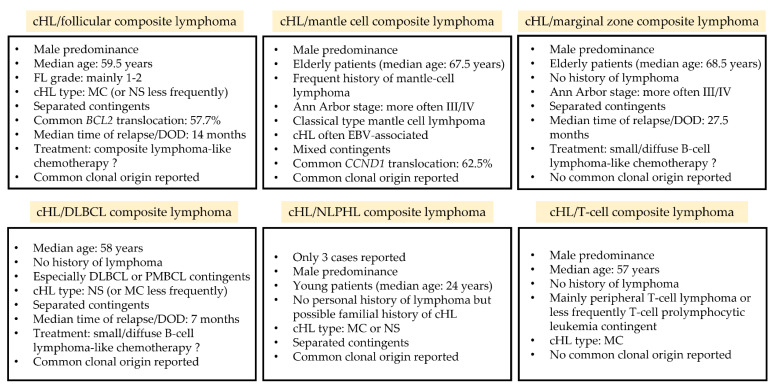
Key features of composite lymphomas associating a cHL with another lymphoma. cHL: classical Hodgkin lymphoma; DOD: death of disease; DLBCL: diffuse large B-cell lymphoma; EBV: Epstein–Barr virus; FL: follicular lymphoma; MC: mixed cellularity; M/F: male/female ratio; NLPHL: nodular lymphocyte-predominant Hodgkin lymphoma; NS: nodular sclerosis; PMBL: primary mediastinal B-cell lymphoma.

**Figure 2 cancers-14-05695-f002:**
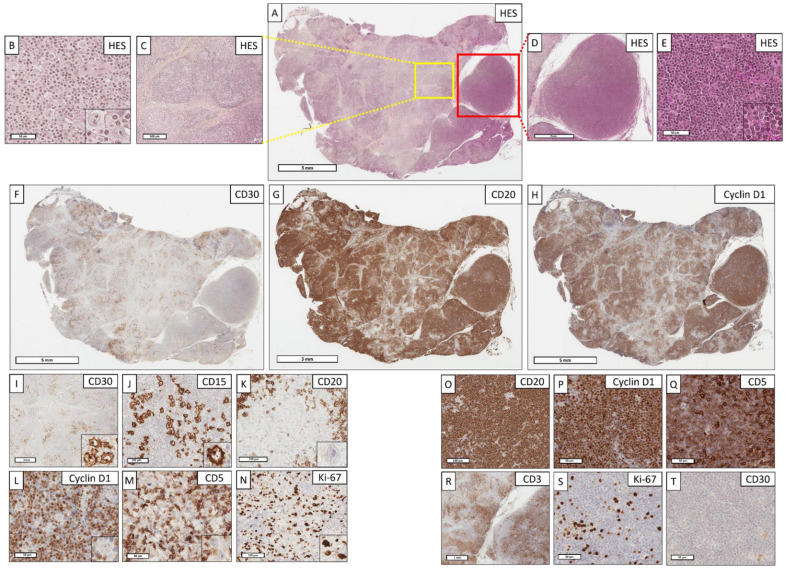
A cHL/mantle cell composite lymphoma with mixed contingents: how to differentiate Hodgkin cells from Hodgkin-like cells? (**A**) (Hematoxylin–eosin–saffron (HES), ×4). Lymph node with a completely remodeled architecture. The tumor proliferation is nodular and diffuse. (**B**) (HES, ×400); (**C**) (HES, ×40, corresponding to yellow box from (**A**). The cHL contingent organization is nodular, delimited by a marked collagenous fibrosis/sclerosis. This contingent is composed of a rich epithelioid histiocytic stroma within which are Hodgkin cells with an atypical nucleus and a large eosinophilic nucleolus. (**D**) (HES, ×16, corresponding to red box from (**A**); (**E**) (HES, ×400). The mantle cell lymphoma contingent organization is diffuse and nodular, consisting of small, slightly atypical cells. (**F**) (CD30, ×4); (**G**) (CD20, ×4) and (**H**) (Cyclin D1, ×4). The anti-CD30 antibody highlights the nodular organization of cHL (localized on the left side of the lymph node), whereas the CD20 and Cyclin D1 antibodies at low magnification highlight the mantle cell lymphoma contingent (predominantly localized on the right side of the lymph node, but also mixed with cHL on the left). (**I**) (CD30, ×20); (**J**) (CD15, ×200); (**K**) (CD20, ×400); (**L**) (Cyclin D1, ×400); (**M**) (CD5, ×400). Hodgkin cells intensely and diffusely express CD30 and CD15, but do not express CD20, Cyclin D1, or CD5. (**N**) (Ki67, ×400). The proliferative index highlights the large Hodgkin cells. (**O**) (CD20, ×200); (**P**) (Cyclin D1, ×400); (**Q**) (CD5, ×400); (**R**) (CD3, ×20); (**S**) (Ki67, ×400); (**T**) (CD30, ×400). The mantle cell lymphoma contingent expresses intensely and diffusely CD20, CD5, and Cyclin D1, without expression of CD3. The tumor cells express CD5 less intensely than adjacent reactive T cells, also expressing CD3. The proliferative index is weak (<10%). In this example, the diagnosis is a cHL/mantle cell composite lymphoma rather than Hodgkin-like cells of a mantle cell lymphoma, since there is a compatible architectural change (sclerotic and nodular), a cHL histiocytic stroma, positive cHL markers (CD30 and CD15) [36,59], and a lack of expression of other markers by the cHL contingent (CD20, CD5, Cyclin D1).

**Figure 3 cancers-14-05695-f003:**
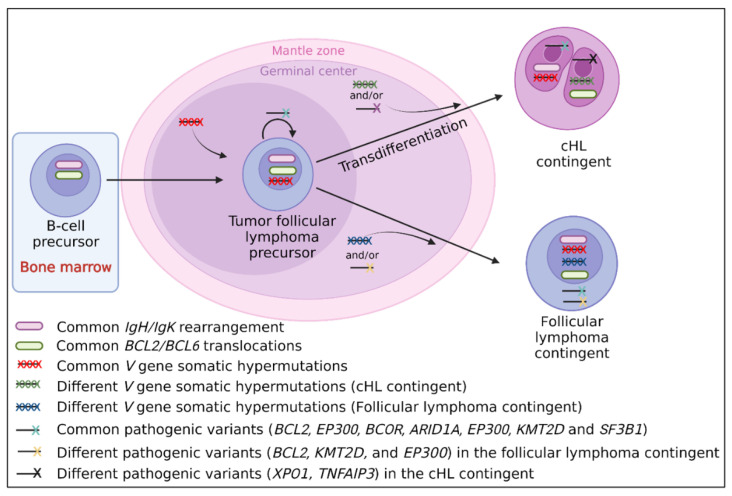
Suggested transdifferentiation model of a follicular lymphoma precursor into classical Hodgkin lymphoma (cHL) contingent in cHL/follicular composite lymphomas. The transdifferentiation model in this type of composite lymphoma is based on the presence of (i) *BCL2/BCL6* translocations in both contingents and similar *IgH/IgK* rearrangements which take place in early B-cells in the bone marrow [10,123]; (ii) identical *V* gene somatic hypermutations [6,11,12], indicating that the tumoral precursor matures through the germinal center; (iii) identical driver pathogenic follicular lymphoma-like variants (*KMT2D, BCL2, EP300*, and *ARID1A*) [10], indicating that the common tumoral precursor is probably a tumor follicular cell; (iv) different *V* gene somatic hypermutations between both contingents [6,11,12], indicating that the separation of both contingents takes place during the germinal center step; and (v) different passenger variants, specific to each contingent (*XPO1, TNFAIP3* found in the cHL contingent only, and *BCL2, KMT2D,* and *EP300* found in the follicular lymphoma contingent only) [10], indicating that each contingent evolves independently after the transdifferentiation step.

**Figure 4 cancers-14-05695-f004:**
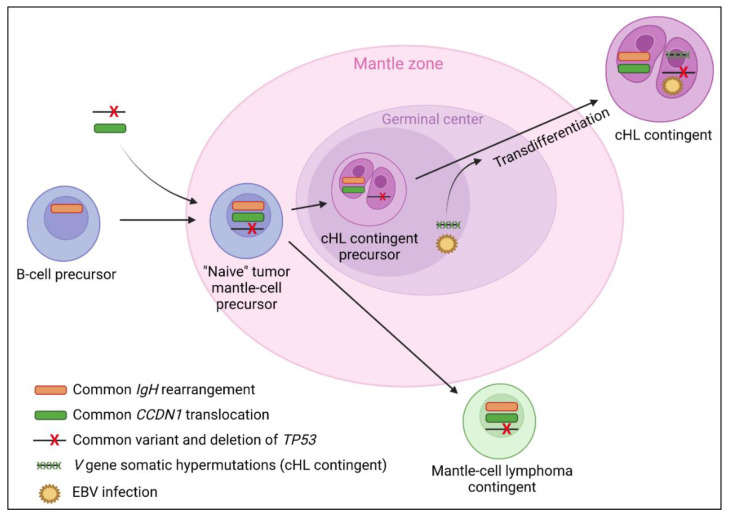
Suggested lymphomagenesis model for classical Hodgkin lymphoma (cHL)/mantle cell composite lymphomas. The frequent presence of a mantle cell lymphoma in patient personal history at diagnosis, *CCND1* translocations in both contingents [6,8,70,71,72], and similar clonal *IgH* rearrangements [8,9] between both contingents suggest a transdifferentiation of cHL from a tumoral mantle cell precursor. One study also reported a common pathogenic variant and deletion of *TP53* shared by both contingents [8]. There is evidence that the cHL contingent could evolve through the germinal center with the presence of *V* gene somatic hypermutations reported in the cHL contingent only [6,9]. The passage through the germinal center of a subclone of the mantle cell lymphoma precursor, with the apparition of *V* gene somatic hypermutations, could be implicated in the transdifferentiation event [9] as could the EBV infection of a tumoral mantle cell precursor, which is very prevalent in the cHL contingent (75%) and never reported in the mantle cell lymphoma contingent. However, 25% of the cHL contingents do not express EBER and/or LMP-1, which suggests that the latter assumption alone is not sufficient to fully explain the lymphomagenesis of the cHL contingent. Although the origin of mantle cell lymphoma from a naive B-cell of the mantle zone is yet to be commonly accepted [123], one study also reported the presence of *V* gene somatic hypermutations in both cHL and mantle cell lymphoma contingents [8], which could suggest a post-germinal center origin of both contingents in some cases.

**Figure 5 cancers-14-05695-f005:**
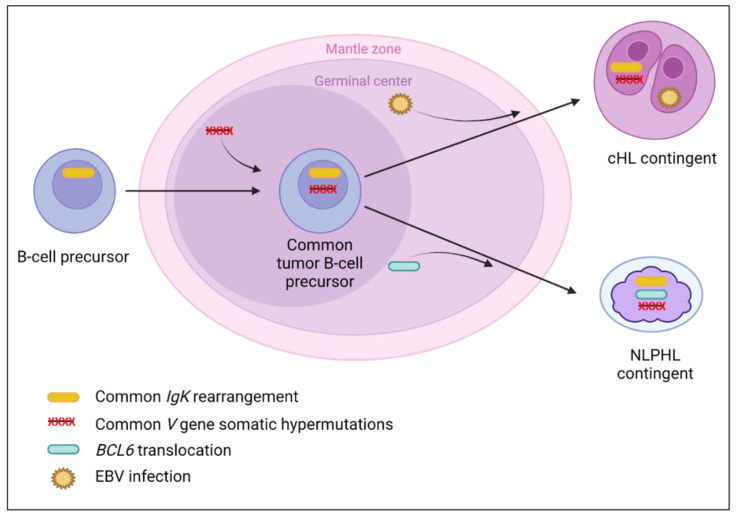
Suggested lymphomagenesis model for classical Hodgkin lymphoma (cHL)/nodular lymphocyte-predominant Hodgkin lymphoma (NLPHL) composite lymphomas. Both contingents of these lymphomas come from the germinal center [123]. The presence of similar clonal rearrangements of *IgK* only with polyclonal *IgH* rearrangements [106], or identical *V* gene somatic hypermutations [107] between both contingents, suggest that the lymphomagenesis of this entity takes place in the germinal center, or after this step. Indeed, the *V* gene somatic hypermutations, arising in the germinal center [123], could interfere with the detection of similar *IgH* rearrangements by avoiding the hybridization of PCR primers on the variable regions. Then, after the germinal center step, additional molecular abnormalities could occur to form both contingents with specific molecular alterations. These abnormalities could be (i) the *BCL6* translocation, which is the hallmark of NLPHL, leading to the BCL6 overexpression reported in two cases, without BCL6 expression in the cHL contingent [106,107]; and/or (ii) EBV infection in cHL, absent in the NLPHL contingent [106]. However, few molecular data are available for this composite lymphoma, which complicates lymphomagenesis understanding.

**Figure 6 cancers-14-05695-f006:**
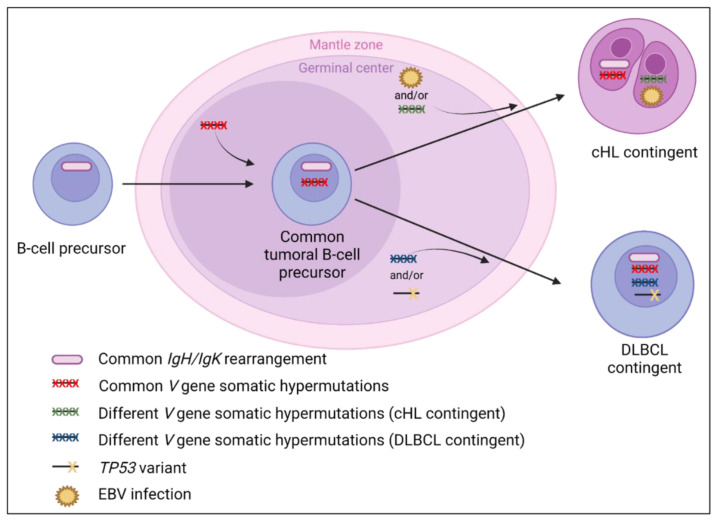
Suggested lymphomagenesis of classical Hodgkin lymphoma (cHL)/diffuse large B-cell lymphoma (DLBCL) composite lymphomas. Similar clonal *IgH*/*IgK* rearrangements between both contingents are reported and support the common clonal origin. Moreover, the presence of common and distinct *V* gene somatic hypermutations [6,13] support that the tumor precursor could originate from the germinal center. Schmitz et al. have hypothesized that a *TP53* variant, identified only in the DLBCL contingent, could contribute to its formation. In contrast, EBV infection is more prevalent in the Hodgkin cells than in the DLBCL contingent (Table 3), which could contribute to the genesis of the cHL contingent in some cases of composite lymphomas, as suggested by Zettl et al. [76]. However, for EBV-negative cHL cases, further molecular studies are needed to explain the lymphomagenesis more precisely.

**Table 1 cancers-14-05695-t001:** Clinical, laboratory, and pathological data.

Clinical, Laboratory, and Pathological Data	cHL/Follicular Lymphoma	cHL/Mantle Cell Lymphoma	cHL/Marginal Zone Lymphoma	cHL/DLBCL	cHL/NLPHL	cHL/T-Cell Lymphoma
**Number of cases included**	76	11	21	79	3	11
**Median age at presentation**	59.5	67.5	68.5	58	24	57
**M/F ratio**	2	4	4	1.5	3/0	2.7
**History of lymphoma, %**						
Yes	7.9 (6/76)	54.5 (6/11)	14.3 (3/21)	6.3 (5/79)	0 (0/3)	27.3 (3/11)
No	59.2 (45/76)	45.5 (5/11)	76.2 (16/21)	57 (45/79)	100 (3/3)	72.7 (8/11)
NA	32.9 (25/76)	0 (0/11)	9.5 (2/21)	36.7 (29/79)	0 (0/3)	0 (0/11)
Location, %						
Unique	6.6 (5/76)	9.1 (1/11)	9.5 (2/21)	10.1 (8/79)	0 (0/3)	9.1 (1/11)
Multiple	34.2 (26/76)	81.8 (9/11)	76.2 (16/21)	48.1 (38/79)	66.7 (2/3)	45.5 (5/11)
NA	59.2 (45/76)	9.1 (1/11)	14.3 (3/21)	41.8 (33/79)	33.3 (1/3)	45.5 (5/11)
**Median size of larger LN/tumor (cm)**	4.15	3	3	3.75	3	3.05
B symptoms, %						
Yes	21 (16/76)	9.1 (1/11)	9.5 (2/21)	11.4 (9/79)	33.3 (1/3)	27.3 (3/11)
No	25 (19/76)	72.7 (8/11)	76.2 (16/21)	26.6 (21/79)	66.7 (2/3)	18.2 (2/11)
NA	53.9 (41/76)	18.2 (2/11)	14.3 (3/21)	62 (49/79)	0 (0/3)	54.5 (6/11)
**LDH dosage, %**						
increased	3.9 (3/76)	9.1 (1/11)	0 (0/21)	5.1 (4/79)	0 (0/3)	9.1 (1/11)
normal range	26.3 (20/76)	9.1 (1/11)	4.8 (1/21)	2.5 (2/79)	0 (0/3)	9.1 (1/11)
NA	69.7 (53/76)	81.8 (9/11)	95.2 (20/21)	92.4 (73/79)	100 (3/3)	81.8 (9/11)
**Ann Arbor stage, %**						
I/II	18.4 (14/76)	27.3 (3/11)	9.5 (2/21)	13.9 (11/79)	33.3 (1/3)	27.3 (3/11)
III/IV	36.8 (28/76)	63.6 (7/11)	66.7 (14/21)	21.5 (17/79)	66.7 (2/3)	45.5 (5/11)
NA	44.7 (34/76)	9.1 (1/11)	23.8 (5/21)	64.6 (51/79)	0 (0/3)	27.3 (3/11)
**BM involvement, %**						
Yes	19.7 (15/76)	27.3 (3/11)	14.3 (3/21)	10.1 (8/79)	0 (0/3)	18.2 (2/11)
No	13.2 (10/76)	0 (0/11)	4.8 (1/21)	2.5 (2/79)	66.7 (2/3)	9.1 (1/11)
NA	67.1 (51/76)	72.7 (8/11)	80.9 (17/21)	87.3 (69/79)	33.3 (1/3)	72.7 (8/11)
**cHL type, %**						
cHL-NS	23.7 (18/76)	9.1 (1/11)	4.8 (1/21)	53.2 (42/79)	33.3 (1/3)	9.1 (1/11)
cHL-MC/granulomatous	60.5 (46/76)	9.1 (1/11)	57.1 (12/21)	21.5 (17/79)	66.7 (2/3)	45.5 (5/11)
cHL-LR	0 (0/76)	18.2 (2/11)	0 (0/21)	1.3 (1/79)	0 (0/3)	0 (0/11)
unclassified	15.8 (12/76)	63.6 (7/11)	38.1 (8/21)	24 (19/79)	0 (0/3)	45.5 (5/11)
**Location of both contingents, %**						
separated	46.1 (35/76)	9.1 (1/11)	42.9 (9/21)	34.2 (27/79)	100 (3/3)	9.1 (1/11)
mixed	19.7 (15/76)	54.5 (6/11)	14.3 (3/21)	0 (0/69)	0 (0/3)	9.1 (1/11)
NA	34.2 (26/76)	36.4 (4/11)	42.9 (9/21)	65.8 (52/79)	0 (0/3)	81.8 (9/11)
Outcome, %						
CR or PR but no relapse (alive)	28.9 (22/76)	18.2 (2/11)	38.1 (8/21)	20.3 (16/79)	66.7 (2/3)	36.4 (4/11)
Relapse and/or DOD	25 (19/76)	9.1 (1/11)	23.8 (5/21)	11.4 (9/79)	0 (0/3)	18.2 (2/11)
Death from other cause	0 (0/76)	18.2 (2/11)	9.5 (2/21)	2.5 (2/79)	0 (0/3)	0 (0/11)
NA	46.1 (35/76)	54.5 (6/11)	28.6 (6/21)	65.8 (52/79)	33.3 (1/3)	45.5 (5/11)
Median follow-up duration, months	23.5	NA	12	12	NA	30
Median time of relapse/DOD, months	14	NA	27.5	7	NA	NA

BM: bone marrow; cHL: classical Hodgkin lymphoma; DLBCL: diffuse large B-cell lymphoma; DOD: death of disease; F: female; LDH: lactate dehydrogenase; LN: lymph node; LR: lymphocyte-rich; M: male; MC: mixed-cellularity; NA: not available; Nb: number; NLPHL: nodular lymphocyte-predominant Hodgkin lymphoma; NS: Nodular sclerosis.

**Table 2 cancers-14-05695-t002:** Treatments for cHL composite lymphomas reported in the literature.

Treatment (First Line)	cHL/Follicular Composite Lymphoma	cHL/Mantle Cell Composite Lymphoma	cHL/Marginal Zone Composite Lymphoma	cHL/DLBCL	cHL/NLPHL	cHL/T-Cell Composite Lymphoma
**Data available, %**	52.6 (40/76)	54.5 (6/11)	81 (17/21)	50.6 (40/79)	66.7 (2/3)	63.6 (7/11)
**cHL-like chemotherapy, %**	25 (10/40)	-	17.6 (3/17)	5 (2/40)	50 (1/2)	42.9 (3/7)
Relapse and/or DOD	50 (5/10) *	-	33.3 (1/3)	50 (1/2)	-	33.3 (1/3)
Alive in remission	50 (5/10)	-	66.7 (2/3)	-	100 (1/1)	66.7 (2/3)
Loss of follow-up/NA	-	-	-	50 (1/2)	-	-
**Small/diffuse B-cell lymphoma-like chemotherapy, %**	30 (12/40)	66.7 (4/6)	47.1 (8/17)	60 (24/40)	-	-
Relapse and/or DOD	41.7 (5/12) **	25 (1/4)	25 (2/8)	16.7 (4/24)	-	-
Alive in remission	41.7 (5/12)	50 (2/4)	62.5 (5/8)	41.7 (10/24)	-	-
Loss of follow-up/NA	16.7 (2/12)	-	12.5 (1/8)	37.5 (9/24)	-	-
Death from another cause	-	25 (1/4)	-	4.2 (1/24)	-	-
**T-cell lymphoma-like chemotherapy, %**	-	-	-	-	-	14.3 (1/7)
Loss of follow-up/NA	-	-	-	-	-	100 (1/1)
**Composite lymphoma-like chemotherapy, %**	35 (14/40)	16.7 (1/6)	11.8 (2/17)	20 (8/40)	-	42.9 (3/7)
Relapse and/or DOD	35.7 (5/14) ***	-	50 (1/2)	37.5 (3/8)	-	-
Alive in remission	57.1 (8/14)	-	-	25 (2/8)	-	66.7 (2/3)
Loss of follow-up/NA	7.1 (1/14)	-	-	37.5 (3/8)	-	-
Death from another cause	-	100 (1/1)	50 (1/2)	-	-	33.3 (1/3)
**Radiotherapy only, %**	2.5 (1/40)	-	11.8 (2/17)	2.5 (1/40)	50 (1/2)	-
Relapse and/or DOD	-	-	-	-	-	-
Alive in remission	100 (1/1)	-	50 (1/2)	100 (1/1)	100 (1/1)	-
Loss of follow-up/NA	-	-	50 (1/2)	-	-	-
**No treatment, %**	7.5 (3/40)	-	-	-	-	-
Relapse and/or DOD	66.7 (2/3)	-	-	-	-	-
Alive in remission	-	-	-	-	-	-
Loss of follow-up/NA	33.3 (1/3)	-	-	-	-	-
**Other treatment, %**	-	16.7 (1/6) ^¥^	11.8 (2/17) ^Ψ^	12.5 (5/40) ^£^	-	-

* Including 3 patients who relapsed on the follicular lymphoma contingent; ** Including 3 patients who relapsed on the cHL contingent; *** Including 2 patients who relapsed on the follicular lymphoma contingent; ^¥^ Prednisone only in one case; ^Ψ^ Gastrectomy in one case and *Helicobacter pylori* eradication in one case; ^£^ Chemotherapy (unspecified) +/− radiotherapy in three cases, gastrectomy in two cases. cHL: classical Hodgkin lymphoma; DOD: death of disease; DLBCL: diffuse large B-cell lymphoma; NA: not available; NLPHL: nodular lymphocyte-predominant Hodgkin lymphoma.

**Table 3 cancers-14-05695-t003:** Immunophenotype of both contingents in cHL composite lymphomas.

Markers	cHL	Follicular Lymphoma	cHL	Mantle Cell Lymphoma	cHL	Marginal Zone Lymphoma	cHL	DLBCL	cHL	NLPHL	cHL	T-Cell Lymphoma
**CD30 **, %**	100 (59/59)	0 (0/21)	100 (10/10)	0 (0/4)	100 (19/19)	6.7 (1/15) *	100 (46/46)	29 (9/31) *	100 (3/3)	33.3 (1/3) *	100 (10/10)	0 (0/4)
**CD15, %**	62.1 (41/66)	0 (0/21)	90 (9/10)	0 (0/4)	73.7 (14/19)	0 (0/14)	81.8 (45/55)	2.4 (1/42) *	100 (3/3)	0 (0/3)	80 (8/10)	0 (0/3)
**CD45/LCA, %**	10.7 (3/28)	100 (24/24)	25 (1/4) *	100 (4/4)	0 (0/16)	92.9 (13/14)	7.7 (2/26)	100 (29/29)	NA	50 (1/2)	0 (0/3)	100 (2/2)
**CD20, %**	33.3 (20/60) *	100 (56/56)	33.3 (3/10) *	100 (9/9)	55.6 (10/18) *	100 (19/19)	27.7 (13/47) *	96.7 (59/61)	50 (1/2) *	100 (2/2)	22.2 (2/9)	NA
**CD79a, %**	42.9 (6/14) *	85.7 (6/7)	33.3 (1/3)	100 (4/4)	7.7 (1/13) *	100 (12/12)	5.9 (1/17)	93.8 (15/16)	NA	NA	NA	NA
**PAX5, %**	97.8 (45/46)	87.5 (7/8)	100 (5/5)	100 (4/4)	66.7 (2/3)	NA	84.6 (11/13)	100 (11/11)	100 (2/2)	100 (2/2)	100 (6/6)	NA
**MUM1, %**	100 (13/13)	22.2 (2/9)	NA	NA	100 (7/7)	57.1 (4/7)	90 (9/10)	75 (8/12)	NA	NA	NA	NA
**OCT2, %**	46.1 (6/13)	80 (4/5)	75 (3/4)	100 (2/2)	0 (0/4)	NA	37.5 (3/8)	100 (5/5)	50 (1/2)	100 (2/2)	NA	NA
**BOB1, %**	38.5 (5/13)	60 (3/5)	33.3 (1/3)	NA	0 (0/4)	NA	28.6 (2/7)	100 (6/6)	NA	NA	NA	NA
**EBER, %**	26.2 (11/42)	0 (0/18)	75 (6/8)	0 (0/5)	58.3 (7/12)	7.1 (1/14)	39.2 (20/51)	29.2 (14/48)	50 (1/2)	0 (0/3)	33.3 (3/9)	0 (0/5)
**LMP-1, %**	NA	NA	100 (4/4)	0 (0/3)	53.3 (8/15)	7.7 (1/13)	54.5 (6/11)	10 (1/10)	NA	NA	NA	NA
**BCL2, %**	48.5 (16/33)	95.8 (46/48)	25 (1/4)	85.7 (6/7)	88.9 (8/9)	100 (11/11)	87.5 (7/8)	75 (9/12)	NA	NA	NA	NA
**BCL6, %**	43.3 (13/30)	100 (37/37)	0 (0/3)	0 (0/5)	NA	NA	20 (1/5)	33.3 (5/15)	0 (0/2)	100 (2/2)	NA	NA
**CD10, %**	6.7 (2/30)	92.7 (38/41)	0 (0/2)	0 (0/5)	NA	NA	0 (0/6)	17.6 (3/17)	NA	NA	NA	0 (0/2)
**Cyclin D1, %**	NA	NA	42.9 (3/7)	100 (10/10)	NA	0 (0/2)	0 (0/2)	NA	NA	NA	NA	NA
**SOX11, %**	NA	NA	50 (1/2)	100 (2/2)	NA	NA	NA	NA	NA	NA	NA	NA
**Kappa light chain, %**	NA	NA	NA	NA	0 (0/11)	33.3 (4/12)	NA	NA	NA	NA	NA	NA
**Lambda light chain, %**	NA	NA	NA	NA	0 (0/11)	16.7 (2/12)	NA	NA	NA	NA	NA	NA
**MAL, %**	NA	NA	NA	NA	NA	NA	0 (0/2)	100 (7/7) ***	NA	NA	NA	NA
**ALK-1, %**	NA	NA	0 (0/2)	NA	NA	0 (0/2)	0 (0/2)	NA	NA	NA	NA	NA
**CD23, %**	NA	NA	0 (0/2)	0 (0/8)	NA	NA	NA	NA	NA	NA	NA	NA
**CD2, %**	NA	NA	NA	NA	NA	NA	NA	NA	NA	NA	NA	100 (2/2)
**CD3, %**	NA	NA	0 (0/3)	0 (0/2)	0 (0/2)	0 (0/2)	0 (0/11)	0 (0/11)	NA	NA	0 (0/3)	100 (11/11)
**CD4, %**	NA	NA	NA	NA	NA	NA	0 (0/2)	NA	NA	NA	NA	50 (4/8)
**CD5, %**	NA	NA	20 (1/5) *	50 (4/8)	0 (0/2)	0 (0/3)	0 (0/4)	0 (0/4)	NA	NA	NA	60 (3/5)
**CD7, %**	NA	NA	NA	NA	NA	NA	NA	NA	NA	NA	NA	33.3 (1/3)
**CD8, %**	NA	NA	NA	NA	NA	NA	NA	NA	NA	NA	NA	66.7 (6/9)
**p53, %**	NA	NA	100 (2/2)	50 (1/2)	100 (7/7)	0 (0/8)	100 (6/6)	16.7 (1/6)	NA	NA	NA	NA
**MYC, %**	NA	NA	NA	NA	NA	NA	NA	100 (2/2)	NA	NA	NA	NA
**TIA1, %**	NA	NA	NA	NA	NA	NA	NA	NA	NA	NA	NA	83.3 (5/6)
**Granzyme B, %**	NA	NA	NA	NA	NA	NA	NA	NA	NA	NA	NA	0 (0/6)
**FOXP3, %**	NA	NA	NA	NA	NA	NA	NA	NA	NA	NA	NA	50 (1/2)

Each immunophenotype of cHL contingent and second contingent (on its right) of composite lymphomas are presented. cHL: classical Hodgkin lymphoma; DLBCL: diffuse large B-cell lymphoma; NA: not available; NLPHL: nodular lymphocyte-predominant Hodgkin lymphoma; * Weak and/or focal expression in most cases; ** Always positive in cHL contingent, as CD30 negativity was an exclusion criterion; *** Primary mediastinal B-cell lymphoma in all cases.

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
