# Peer review of "Plasticity in Classical Hodgkin Composite Lymphomas: A Systematic Review"

_cancers, 2022, doi:10.3390/cancers14225695_

Round 1
Reviewer 1 Report
This is a very timely review of rare and diagnostically and clinically challenging composite lymphomas. The review follows the PRISM guidelines and is well done.
I think the title is misleading- in that not all the cases reviewed are worked up to the "a state of the art" level. I would suggest using "a timely review" or "comprehensive review.
In addition, the text is quite dense to read- and would benefit from the inclusion of a few figures that capture some of the key features of at some of the composite lymphomas.
Author Response
Response to the comments of the reviewers
First, we would like to thank all the reviewers and the editors for their comments that have helped improve the quality of the manuscript and for the time they dedicated towards reading our work. Regarding the comments added in the main text by the proofreader "MDPI": we have added a line in the methods section to confirm that all figures included in this review are original (page 4, lines 43 to 45). However, despite our efforts to register the review in PROSPERO, we do not have a registration number yet. We will get back to you as soon as we have more information.
Reviewer #3
Comment #1:
This is a very timely review of rare and diagnostically and clinically challenging composite lymphomas. The review follows the PRISM guidelines and is well done. I think the title is misleading- in that not all the cases reviewed are worked up to the "a state of the art" level. I would suggest using "a timely review" or "comprehensive review.
Answer#1:
As suggested by this reviewer and reviewer #2, the title was misleading and has been changed to comply with the PRISMA 2020 checklist guidelines. The revised title now reads: “Plasticity in classical Hodgkin composite lymphomas: a systematic review”.
Comment #2:
In addition, the text is quite dense to read- and would benefit from the inclusion of a few figures that capture some of the key features of at some of the composite lymphomas.
Answer#2:
The comment of this reviewer is quite relevant, since the results are indeed dense. A figure summarizing the main key points has been added (Figure 1, page 39).
Reviewer 2 Report
This review summarized the clinical, histopathological, immunohistochemical, and molecular data for each composite lymphoma. The challenge for pathologist is to differentiate real Hodgkin-cells of cHL from Hodgkin-like cells associated with a non-Hodgkin lymphoma. The language is not well. The comments are as below:
1、 P4, Line 5: the range of publications year should be added;
2、 P6, Section 2.3: The paragraph should be aligned in parallel;
3、 P7, Line 2: remove “at presentation”
4、 P8,The sentence “The cHL contingent was of NS-type in 33.3% (1/3) and MC-type in
66.7% (2/3) of cases.” can be incorporated into the next paragraph.
5、 P12, Line 32: revise detected to observed
Author Response
Response to the comments of the reviewers
First, we would like to thank all the reviewers and the editors for their comments that have helped improve the quality of the manuscript and for the time they dedicated towards reading our work. Regarding the comments added in the main text by the proofreader "MDPI": we have added a line in the methods section to confirm that all figures included in this review are original (page 4, lines 43 to 45). However, despite our efforts to register the review in PROSPERO, we do not have a registration number yet. We will get back to you as soon as we have more information.
Reviewer #1
This review summarized the clinical, histopathological, immunohistochemical, and molecular data for each composite lymphoma. The challenge for pathologist is to differentiate real Hodgkin-cells of cHL from Hodgkin-like cells associated with a non-Hodgkin lymphoma. The language is not well.
Comment #1:
P4, Line 5: the range of publications year should be added;
Answer#1:
This has been modified in the manuscript: “without limit regarding the year of publication (from 1977 to 2022)” (page 4, line 6). Moreover, the manuscript has been revised for English language by a medical writer.
Comment #2:
P6, Section 2.3: The paragraph should be aligned in parallel.
Answer#2:
This has been modified in the manuscript (page 6, section 2.3).
Comment #3:
P7, Line 2: remove “at presentation”
Answer#3:
This has been modified in the manuscript: “Most patients (76.2%, 16/21) had no history of lymphoma” (page 7, line 2).
Comment #4:
P8, The sentence “The cHL contingent was of NS-type in 33.3% (1/3) and MC-type in 66.7% (2/3) of cases.” can be incorporated into the next paragraph.
Answer#4:
This has been modified accordingly in the manuscript (page 8, line 21).
Comment #5:
P12, Line 32: revise detected to observed
Answer#5:
This has been modified accordingly in the manuscript: “In addition, no shared B or T rearrangement were observed in the only two cases for which the contingents were individualized” (page 12, line 47).
Reviewer 3 Report
In this manuscript Dr Trecourt et al perform a systematic review of the literature to identify composite lymphomas and describe their characteristics. This is excellent work and I congratulate the authors on this. However I have several recommendations:
1. Firstly, it was not immediately obvious that this was a systematic review. Based on PRISMA guidelines, this needs to be included in the title. I did review the authors PRISMA checklist and they do note that this is done - however the title that I have for this manuscript does not indicate a systematic review: "Plasticity in classical Hodgkin composite lymphomas:state of the art".
2. The authors report throughout on the common pathophysiology of some cHL cases occurring in composite with B-cell lymphomas based on expression of GCB markers. This is provocative. For many decades it has been known that lymphoma pathophysiology follows the lymphoma negation principle. An indolent GCB lymphoma will transform into an aggressive GCB lymphoma. eg. a FL will transform into a GCB DLBCL with CD10 positivity, etc. Similarly indolent post-centric lymphomas (activated-B cell lymphoma) will transform to aggressive ABC lymphomas., eg. an LPL can rarely transform into an ABC DLBCL. How do the author's data fit into this context and how are they able to explain the co-existence of GGCB and ABC lymphomas from a pathophysiological perspective? Its very important to talk about the lymphoma negation principle in this context. I refer the authors to an old but very helpful review on the topic by Leonard Tan: PMID: 19404843. In fact, even EBV positive GCB and ABC lymphomas, when they do co-occur, follow different pathophysiological mechanisms.
Thank you for the opportunity to review this work.
Author Response
Response to the comments of the reviewers
First, we would like to thank all the reviewers and the editors for their comments that have helped improve the quality of the manuscript and for the time they dedicated towards reading our work. Regarding the comments added in the main text by the proofreader "MDPI": we have added a line in the methods section to confirm that all figures included in this review are original (page 4, lines 43 to 45). However, despite our efforts to register the review in PROSPERO, we do not have a registration number yet. We will get back to you as soon as we have more information.
Reviewer #2
In this manuscript Dr Trecourt et al perform a systematic review of the literature to identify composite lymphomas and describe their characteristics. This is excellent work and I congratulate the authors on this.
Comment #1:
However I have several recommendations: 1. Firstly, it was not immediately obvious that this was a systematic review. Based on PRISMA guidelines, this needs to be included in the title. I did review the authors PRISMA checklist and they do note that this is done - however the title that I have for this manuscript does not indicate a systematic review: "Plasticity in classical Hodgkin composite lymphomas:state of the art".
Answer#1:
As suggested by this reviewer and reviewer #3, the title was misleading and has been changed to comply with the PRISMA 2020 checklist guidelines. The revised title now reads: “Plasticity in classical Hodgkin composite lymphomas: a systematic review”.
Comment #2:
- The authors report throughout on the common pathophysiology of some cHL cases occurring in composite with B-cell lymphomas based on expression of GCB markers. This is provocative. For many decades it has been known that lymphoma pathophysiology follows the lymphoma negation principle. An indolent GCB lymphoma will transform into an aggressive GCB lymphoma. eg. a FL will transform into a GCB DLBCL with CD10 positivity, etc. Similarly indolent post-centric lymphomas (activated-B cell lymphoma) will transform to aggressive ABC lymphomas., eg. an LPL can rarely transform into an ABC DLBCL. How do the author's data fit into this context and how are they able to explain the co-existence of GGCB and ABC lymphomas from a pathophysiological perspective? Its very important to talk about the lymphoma negation principle in this context. I refer the authors to an old but very helpful review on the topic by Leonard Tan: PMID: 19404843. In fact, even EBV positive GCB and ABC lymphomas, when they do co-occur, follow different pathophysiological mechanisms. Thank you for the opportunity to review this work.
Answer#2:
We thank the reviewer for raising this key point, which represents a particularly difficult issue. The classification of lymphomas based on the pathophysiology and the origin of the tumor cell is classically accepted, but contrasts with pathologies such as composite lymphomas, for which both contingents often share a common origin. Classical Hodgkin's lymphoma could thus have a particular pathophysiology when it occurs within composite lymphomas. In the discussion section, we suggest that this is the case for composite cHL/follicular lymphomas, for which there are morphological (frequent EBV-negative mixed cellularity type, abundant histiocytic stroma), immunohistochemical (expression of BCL2, BCL6 by Hodgkin cells), cytogenetic (translocation of BCL2 or BCL6 in both contingents), and molecular differences (same IgH, IgK rearrangements, same "follicular lymphoma-like" pathogenic variant of both contingents) compared to Hodgkin's lymphoma arising de novo. Moreover, in recent years, studies have reported the transdifferentiation of certain lymphomas into myeloid neoplasms (e.g., follicular lymphoma and histiocytic/dendritic cell sarcoma sharing a common B-cell origin). Similarly, there are reports of follicular lymphoma transforming into diffuse large B-cell lymphoma with a post-germinative center phenotype, questioning in some rare cases the typical and commonly accepted models of lymphomagenesis. However, these are rare cases, and we must remain cautious, as suggested by the reviewer.
The lymphoma principle of negation is indeed an important and interesting element for the differential diagnosis between a lymphoma and a reactive feature, especially in the context of Hodgkin-like cells. Therefore, we have added two paragraphs to the discussion section of the revised version of the manuscript:
- “Although the classification of lymphomas, and especially the distinction between germinal center and post-germinal center subgroups, is currently based on cellular origin [120], it appears that the cellular plasticity occurring during lymphomagenesis might be higher than initially thought. Follicular lymphomas, which have a germinal center origin [120], are an ideal illustration of this cellular plasticity, since they have been reported to transform into post-germinal center DLBCL [121, 122] or transdifferentiate into myeloid pathologies such as histiocytic/dendritic cell sarcomas in rare instances [123]. The data presented herein thus expand the spectrum of mature B-cell plasticity and the limits of lymphomegenesis.” (page 10, lines 25 to 35).
- “Moreover, it is classically accepted that the lymphoma pathophysiology follows the lymphoma negation principle [120]; thus, the presence of reactive germinal centers and/or post-germinal center immunoblasts with a normal differentiation spectrum are not suggestive of cHL diagnosis [120].” (page 10, lines 45 to 49).
We thank the reviewer for pointing out this element, allowing us to improve the clarity of the manuscript and deepen the discussion.